# Hybrid Breeding for MLN Resistance: Heterosis, Combining Ability, and Hybrid Prediction

**DOI:** 10.3390/plants9040468

**Published:** 2020-04-08

**Authors:** Christine Nyaga, Manje Gowda, Yoseph Beyene, Wilson T. Murithi, Juan Burgueno, Fernando Toledo, Dan Makumbi, Michael S. Olsen, Biswanath Das, Suresh L. M., Jumbo M. Bright, Boddupalli M. Prasanna

**Affiliations:** 1Department of Agricultural Science and Technology, Kenyatta University, Nairobi 43844-00100, Kenya; christinenyaga96@gmail.com (C.N.); wtmuriithi@gmail.com (W.T.M.); 2International Maize and Wheat Improvement Centre (CIMMYT), World Agroforestry Centre (ICRAF), United Nations Avenue, Gigiri, Nairobi 1041-00621, Kenya; Y.Beyene@cgiar.org (Y.B.); d.makumbi@cgiar.org (D.M.); M.Olsen@cgiar.org (M.S.O.); b.das@cgiar.org (B.D.); l.m.suresh@cgiar.org (S.L.M.); b.jumbo@cgiar.org (J.M.B.); b.m.prasanna@cgiar.org (B.M.P.); 3Biometrics and Statistics Unit, International Maize and Wheat Improvement Center (CIMMYT), CarreteraMexico-Veracruz Km. 45, El Batán, Texcoco Estado de Mexico 56237, Mexico; J.Burgueno@cgiar.org (J.B.); F.TOLEDO@cgiar.org (F.T.)

**Keywords:** maize lethal necrosis, combining ability, heterosis, genomic prediction, resistance

## Abstract

Prior knowledge on heterosis and quantitative genetic parameters on maize lethal necrosis (MLN) can help the breeders to develop numerous resistant or tolerant hybrids with optimum resources. Our objectives were to (1) estimate the quantitative genetic parameters for MLN disease severity, (2) investigate the efficiency of the prediction of hybrid performance based on parental *per se* and general combining ability (GCA) effects, and (3) examine the potential of hybrid prediction for MLN resistance or tolerance based on markers. Fifty elite maize inbred lines were selected based on their response to MLN under artificial inoculation. Crosses were made in a half diallel mating design to produce 307 F1 hybrids. All hybrids were evaluated in MLN quarantine facility in Naivasha, Kenya for two seasons under artificial inoculation. All 50 inbreds were genotyped with genotyping-by-sequencing (GBS) SNPs. The phenotypic variation was significant for all traits and the heritability was moderate to high. We observed that hybrids were superior to the mean performance of the parents for disease severity (−14.57%) and area under disease progress curve (AUDPC) (14.9%). Correlations were significant and moderate between line *per se* and GCA; and mean of parental value with hybrid performance for both disease severity and AUDPC value. Very low and negative correlation was observed between parental lines marker based genetic distance and heterosis. Nevertheless, the correlation of GCA effects was very high with hybrid performance which can suggests as a good predictor of MLN resistance. Genomic prediction of hybrid performance for MLN is high for both traits. We therefore conclude that there is potential for prediction of hybrid performance for MLN. Overall, the estimated quantitative genetic parameters suggest that through targeted approach, it is possible to develop outstanding lines and hybrids for MLN resistance.

## 1. Introduction

Maize lethal necrosis (MLN) caused by the co-infection of two viruses, Maize chlorotic mottle virus (MCMV) belonging to the *Tombusviridae* group and any member virus from the *Potyviridae* group mostly Sugarcane mosaic virus (SCMV) is one of the most important maize diseases in sub Saharan Africa (SSA) [1]. MLN has been reported to cause up to 100% yield loss making it a serious threat to food security in SSA [2]. Breeding cultivars for MLN resistance is the most economical and environmental friendly way of controlling MLN [3]. Previous studies revealed that MLN disease resistance is controlled by few major effect and several minor effect genes [4,5]. In this scenario, finding appropriate parental combinations which can carry most of these resistance genomic regions is critical in order to have MLN resistant hybrid. Screening large number of parental lines and selecting appropriate parents for crossing and evaluating them in multiple locations is laborious. Further, the efficiency of finding MLN resistant hybrids largely depends on how competently superior hybrid combinations are identified with optimum use of available inputs. Therefore, understanding the nature of inheritance of MLN resistance and testing of different prediction methods can maximize the chance to develop MLN resistance or tolerance hybrids while using minimal resources.

Hybrid breeding is a gifted approach to enhance crop productivity and yield stability. The development of hybrids involves selection of inbred parents with desired attributes and good combining ability. With the availability of doubled haploid (DH) technology in maize, breeders have the capacity to quickly and accurately create large number of inbred lines with high level of homozygosity [6]. Consequently, millions of potential cross combinations are possible to create but unfortunately, only a small portion of these combinations are possible to evaluate in the field due to limited budget. Thus, identifying the most promising hybrids for field evaluation is cost effective for breeders.

Hybrid performance prediction involves the estimation of the breeding value of the crosses. The parental inbreds and a few crosses are evaluated in the field ensuing the prediction of the untested crosses using the phenotypic data from the tested crosses and genotypic data of the parental inbreds. Various methods have been used to predict the hybrid performance depending on the types of hybrids (single cross or three-way cross) and traits in consideration [7,8]. Selection of best performing elite lines as parents is common in hybrid breeding, however, for complex traits, line *per se* is a poor predictor of hybrid performance, as a result *per se* based predictions are used in simple inherited traits which are not largely affected by non-additive effects [7]. Mid-parent value is another simple predictor of hybrid performance, however, the correlations between mid-parent value and hybrid performance are generally affected by dominance effects revealed in heterosis and imperfect heritability [9]. Prediction of F1 hybrid performance based on general combining ability (GCA) is of another interest and also a promising phenotypic based approach [10]. In this case, the predominance of GCA variance over specific combining ability (SCA) variance is of crucial importance to have an accurate prediction [10]. Further, prior information on GCA effects of line/s is required in order to use GCA based predictions. 

Prediction based on genetic distances between the parents has also been used on the basis that there is a linear relationship between hybrid vigor and the genetic distances between the parents when all QTLs associated with the trait are considered [11]. Best linear unbiased prediction (BLUP) is also used in the prediction of untested single crosses performance using information on genetic relationships among inbreds and phenotypic data of related tested hybrids [12]. Genetic relationships are obtained from either pedigree information or molecular marker data whereby coefficients of coancestry are estimated to measure the degree of similarity [11]. BLUP approach has been applied successfully to predict hybrid performance in maize [13], sunflower [7], and triticale [14].

A vast number of molecular markers are available allowing breeders to use markers in plant breeding programs [15]. With the aim of saving resources, selection of inbred lines either based on GCA or by using molecular markers are paramount. Marker assisted selection (MAS) utilizes only markers with significant association to the loci [16]. The use of only significant associations, however, has limited success in predicting or improving maize hybrid performance. This is mainly because the detected quantitative trait loci (QTL) are specific to a particular genetic background and/or have small additive effects [17]. 

Genomic prediction (GP) in comparison to GCA based prediction, prior information on GCA effects of parental lines is not required, as it uses a model trained on both phenotypic and genotypic data on hybrids without essentially including all hybrid parents [18]. GCA based prediction has a limitation when there is low genetic variance or high SCA variance for the trait of interest. GP contrary to the traditional QTL mapping utilizes genome-wide markers to estimate the effects of all loci and thereby compute a genomic estimated breeding value (GEBV) [13]. The GEBV is therefore the sum of all marker effects and the prediction accuracy is the correlation between the true breeding value and the GEBV. Various approaches utilizing GP to predict hybrid performance have been proposed, including ridge regression-BLUPs [19], Bayesian methods, genomic BLUP (GBLUP) and machine learning [20]. GP has been applied in prediction of hybrid performance for grain yield in maize [11,21], triticale [14], and sunflower [7]. These studies showed the potential of GP in prediction of hybrid performance for important traits.

Plant breeders commonly use diallel mating design to determine the basis of complex traits inheritance and to identify the superior general and specific combiners. In this study we used phenotypic and molecular data of half diallel crosses and parental lines gathered in CIMMYT maize hybrid breeding program and compared several approaches to investigate their potential to predict single-cross performance for MLN resistance. In particular, the objectives of our study were to (1) estimate the variance components for MLN resistance or tolerance (2) assess the extent of mid-parent and better parent heterosis, and (3) predict the performance of F1 hybrids for MLN resistance or tolerance based on mean of the parents, GCA effects, genetic distance, and GP. 

## 2. Results 

### 2.1. Phenotypic Variation

The mean performance of parental lines and F1 hybrids across environments for MLN disease severity (DS) and area under disease progress curve (AUDPC) was 6.25 and 135.28; 5.05 and 136.50, respectively (Table 1). For both parents and hybrids, analysis of variance (ANOVA) revealed highly significant genotypic variance (σ^2^_G_) and GxE (genotype x environment) interaction variances for MLN DS and AUDPC across environments (Table 1). This ruled out the possibility of bias due to environment-specific disease responses in the combined analysis. The variance for GCA (σ^2^_GCA_) and SCA (σ^2^_SCA_) and their interactions with environment were also significant for both MLN DS and AUDPC values (Table 1). For MLN DS and AUDPC values, we observed that relative to the total genotypic variance (σ^2^_G_), the σ^2^_SCA_ estimates amounted to 21% and 22%, respectively. Estimate of heritability for parental lines was 0.42 and 0.86 for MLN DS and AUDPC values, respectively. Whereas for hybrids the estimate of heritability was 0.77 and 0.83 for MLN DS and AUDPC, respectively. The frequency of the phenotypic values in both MLN DS and AUDPC revealed an approximate normal distribution (Figure 1) showing adequate distribution from tolerant to susceptible lines across environments.

The mean values of MLN DS of parents ranged from 3.44 (CLRCY039) to 7.12 (DTPYC9_F13_2_3_1_2_B). Among the 50 parental lines, five lines (CLRCY039, CLRCY034, CLYN231, DTPYC9_F46_1_2_1_2_B, and CLYN261) had scores of < 5 while the highest disease severity values (> 7) were scored from parents CZL068, DTPYC9_F13_2_3_1_2_B, CML544 and CML444. In the case of F_1_ hybrids, around 12 hybrids had scores of <4 and 61 hybrids showed <4.5 score as well as 135 hybrids were showed MLN DS value of <5. The highest and the lowest MLN DS were recorded as 3.5 (CML550 x CML494) and 7.12 (CML443 x CKL5005), respectively. Out of 307 crosses, only 14 hybrids showed MLN DS score of >6.0.

### 2.2. Heterosis and the Genetic Distance among Parental Lines

For MLN DS, negative values for heterosis are preferable as the goal was to develop MLN resistant or tolerant hybrids and the resistance or tolerance is associated with lower values in the 1–9 scale (Table 2). Compared to the mid-parent performance, hybrids had on an average low susceptibility or high resistance or tolerance level to MLN as revealed for DS (−14.57%). Mid-parent heterosis for MLN DS ranged from −38.63% to 43.83%. Compared to better resistant parent, hybrids had an average −04.28% lower susceptibility to MLN. For AUDPC the mean mid-parent and better parent heterosis was 14.19% and 51.19%, respectively.

The Rogers’ genetic distance estimates based on SNP marker data revealed a high variation in relatedness among the parental lines. The distances among the 50 parental lines ranged considerably from zero to 0.54 with a mean of 0.37. In the 50 pairwise comparisons, only 2.6% had genetic distances <0.10. Most of the pairs of parents (40.1%) fell between 0.40 and 0.50. The principal component analyses (PCA) showed clustering among the inbred lines (Figure 2). The first two principal components (PCs) explained 56.6% of the total variation.

The correlation between the line *per se* and the GCA effects of the lines was moderate and significant with r = 0.72 (*p* < 0.01) and 0.58 (*p* < 0.01) for MLN DS and AUDPC, respectively (Figure 3). The correlation between the mid parent heterosis (MPH) and the Rogers’ genetic distance was low and negative for both the traits (Table 2). Mid parent performance was significantly (*p* < 0.05) correlated with F1 hybrid performance for MLN DS (r = 0.29) and AUDPC values (r = 0.54, Figure 4). Compared to predictions based on mid parent values, the GCA based correlations for F1 hybrid performance was higher for both the traits with r = 0.94 (*p* < 0.01) for MLN DS and r = 0.93 (*p* < 0.01) for AUDPC values (Figure 4). Genomic predictions revealed high correlations with hybrid performance, for MLN DS r = 0.74 and for AUDPC values r = 0.77 (Figure 5). 

## 3. Discussion

MLN is a complex disease as the interaction between the two viruses (SCMV and MCMV) and their interaction the with environment are critical for its widespread in the field [4]. Early discovery studies clearly showed the contribution of many genomic regions for MLN resistance [4]. Genetic studies on MLN showed both recessive [22] and dominance [23] type of inheritance in different populations. Predominance of additive effects in the control of MLN resistance or tolerance was also evident in a diallel study which involved 344 F_1_ hybrids [8]. Combining all the favorable alleles from diverse sources of MLN resistance or tolerance into one hybrid combination is challenging. Nevertheless, understanding the genetic differences and the type of gene action involved in MLN resistance or tolerance is useful in prioritizing inbred lines to be used as parents or in hybrid development.

Breeding for MLN resistance is an important goal in SSA in the effort to reduce yield losses and ensure food security [2]. Finding appropriate combinations to form hybrids which contribute to both MLN resistance or tolerance and grain yield is the final goal. Hybrid breeding mainly aims at identifying inbred parents with high genetic diversity and strong heterosis in the F1 generation [24]. In the present study 307 hybrids and their 50 parental lines were evaluated for their response to MLN and the 50 parents were genotyped in order to compare different approaches to predict F1 hybrid performance and investigate the relationship between the mid parent heterosis and the parental genetic distances. 

The genetic variances were significant (*p* < 0.05) for both MLN DS and AUDPC values with moderate to high heritability. This was consistent with previous studies on MLN resistance that observed significant genetic variances and moderate to high heritability [4,5,8,23,25]. GCA and SCA estimates provide useful information on potential parental value which helps the breeders to choose appropriate breeding parents. The significance of the GCA effects indicates that at least one parent differs from other parents in terms of number of favorable genes or alleles. GCA represents additive genetic effects, with recurrent selection, it is possible to accumulate those favorable genes. In the current study for MLN resistance GCA estimates are five times higher than SCA estimates, indicating the importance of additive gene action for MLN resistance. Earlier study also revealed 2–3 times higher GCA estimates over SCA estimates for MLN resistance [7]. Nevertheless, we also observed significant SCA effects which indicating that one cannot completely ignore the non-additive effects, and this can also be exploited to develop better MLN-resistant hybrids. The interaction of both GCA and SCA with the environment was significant (*p* < 0.01) showing that both additive and non-additive genetic effects were influenced by the environment. Higher heritabilities show the amenability of the trait to improve for MLN resistance or tolerance through recurrent selection.

Average genetic distance reported was 0.37 which is comparable with the earlier findings [26,27,28]. Masuka et al., [26] reported distances ranging from 0.004 to 0.40 with a mean of 0.294 from the analysis of 53 parental lines and two F_2_ populations derived from the CIMMYT Africa breeding program. Beyene et al., [27] analyzed the genetic distances of 703 doubled haploids and reported distances ranging from 0.070 to 0.457 with an average of 0.355. Ertiro et al., [28] reported genetic distances which ranged from 0.011 to 0.346 with an average of 0.313 across pairwise combinations of 265 inbred lines. For Beyene et al., [27] most of the distances fell between 0.300 to 0.475 with a frequency of 69% and in addition, less than 5% of the distances fell below 0.10. Masuka et al., [26] on the others reported 97.7% of the pairwise comparisons falling between 0.20 and 0.40 with only 0.5% showing <0.10 genetic distance. Studies however, from different geographical regions have reported relatively contrasting genetic distance like Li et al., [29] who evaluated popcorn lines from China using simple sequence repeats (SSRs) and reported genetic distances ranging from 0.125 to 0.730 with an average of 0.477. These differences could be contributed by different factors such as the type of markers used in the study in that SSRs may show higher distances compared to GBS which covers the whole genome and captures information that might have been missed by SSRs [26]. Other factors include the use of different materials and breeding objectives such as target quality.

Negative and low magnitude of correlations were observed between genetic distance and mid parent heterosis for both MLN DS and AUDPC (Table 2). Negative correlation implies that the higher the genetic distance, the higher the MLN resistance or tolerance however, the magnitude of correlation was low for both DS and AUDPC (Table 2). Previous studies have reported similar results with low but significant negative correlations [10,14,30,31]. Gowda et al., [14] reported no significant association between Rogers’ distance and heterosis from a weakly related panel of triticale lines. Various explanations have been suggested in the relationship between heterosis and genetic distance. This low correlation can be attributed to poor association between heterozygosity estimated from marker data and with one at QTL in the crosses; poor association between heterozygosity and heterosis at the QTL in the crosses; existence of multiple alleles and epistasis [32]. Melchinger et al., [33] explained that heterosis could be predicted from the genetic distance depending on the type of germplasm used and when it is smaller than a certain threshold. 

In hybrid breeding, new lines are chosen as parental lines based on their line *per se* performance. For simple inherited traits, mid parent value can serve as a good predictor of hybrid performance in many crops including maize. For instance, several studies reported moderate to high accuracy of >0.60 between mid-parent value and hybrid performance in maize for days to silking, ear dry matter content, and plant height [34], also in wheat and triticale for plant height, heading time, and 1000-kernel weight [14,35,36,37]. In contrast to this expectation, we observed low to moderate correlations between mid-parent value and hybrid performance for both traits (r = 0.29 for DS and r = 0.54 for AUDPC). This is quite unexpected as we observed predominance of additive effects for MLN (Table 1), the interaction between two viruses SCMV and MCMV and their interaction with environments independently and/or together which is difficult to capture with current study may play a role here in the observed low correlations which warrants further study. The correlation between the mid parent value and the hybrid performance is anticipated to be low in complex traits due to masking dominance effects [38]. MLN is a complex trait because of two viruses and controlled by few major and many minor genes distributed across the genome [4,5,23,25]. Similar low correlations were also observed in several studies for grain yield in maize [11,34,39], wheat [35,37], triticale [14,36,40], sunflower [7], and barley [41]. For less complex traits like heading time and plant height, Gowda et al., [14] reported higher accuracies (r = 0.81 and 0.88 respectively). Nonetheless prediction of hybrid performance based on mid parental value is not high enough to use solely on MLN resistant hybrid prediction however, its positive correlation with hybrid performance is a good indicator to select MLN resistant or tolerant lines as parents.

Availability of prior information on GCA values of selected lines on trait of interest can help the breeders to choose the specific lines for the hybrid combination which can enhance the probabilities of finding best hybrid combinations. Lines CML550, CML494, and CML343 showed low GCA values (<4.5) and low MLN DS values (<5.7) indicating the high chances of success for selecting good GCA lines based on *per se* performance (Figure 3). The observed moderate and significant positive correlation between line *per se* performance and GCA effects showed that the inbreds *per se* performance information can be used to prioritize the parental lines to make hybrid combinations. These results are comparable with past studies whereby Dagne et al., [42] reported highly significant correlations between GCA effects and *per se* performance for gray leaf spot and grain yield. Miedaner et al., [3] also reported highly significant correlation between line per se and GCA effects in wheat.

In hybrid breeding, for the traits governed predominantly by GCA variance over SCA variance, finding superior hybrids is moderately successful with GCA based predictions. For instance, simple inherited traits like grain dry matter content in maize [43], plant height, and heading time in wheat [10,14,35], resistance to leaf rust, powdery mildew and *Septoria tritici* blotch in wheat [35], and oil content in sunflower [7] were effectively predicted for best hybrid combinations based on their GCA effects. Similarly, even though MLN is relatively more complex, GCA is seems to be very effective predictor of best hybrid combination/s which is clear with observed prediction accuracy of >0.90 for both MLN DS and AUDPC values (Figure 4). This is also well supported from the previous study results on GCA based prediction of MLN DS with 344 hybrids [7]. Further, Miedaner et al., [3] also reported high prediction accuracy for hybrid performance of fusarium head blight based on GCA effects (r = 0.86). Dagne et al., [42] also reported high correlation between the sum of GCA values of the parents and the GLS severity of their progeny in maize. Therefore, though SCA effects are important in hybrid breeding for MLN resistance or tolerance, GCA alone can be used as an effective predictor to find the best possible combinations of MLN resistance or tolerance. 

Though prediction based on GCA is effective for MLN resistance, having prior information on GCA effects for all the available lines to be used in hybrid breeding is time consuming and labor intensive. Alternative to phenotypic based GCA prediction, GP models are used to predict the hybrid performance and estimate GCA effects with reduced field evaluation [21,44,45]. With GBLUP model, we observed high prediction accuracy for both DS and AUDPC value (Figure 5). Similar results were also reported in maize with accuracy ranging from 0.60 to 0.98 for several simple inherited traits like grain moisture [13,21], grain dry matter content [44], plant height, days to silking, lignin content, starch content, sugar content [45], root lodging, stock lodging [13], and northern corn leaf blight resistance [46]. GP is also promising for complex traits like grain yield and dry matter yield in maize which predicted with high accuracies ranged from 0.50 to 0.95 [21,44,45]. In conclusion, marker-based prediction is also promising for both simple and complex inherited traits however their accuracy is influenced by marker density, size of tested hybrids and relatedness of tested and untested hybrids.

In classical way of hybrid breeding, forming large number of crosses and further testing them in yield trials is very much time and supply demanding. Therefore, optimally allocating the available resources with increase in selection gain and the probability to identify the hybrids with superior performance is crucial in hybrid breeding. Having prior knowledge on genetic architecture of the trait under evaluation, analysis of genetic variance, estimate of heritability, combining ability, and trait linked markers assist the breeder to predict the untested hybrids performance with more precision. Further predominance of variance due to GCA over SCA effects suggests high recurrent selection gain is possible in hybrid breeding for MLN resistance. GCA tests are feasible but very tedious. Alternatively, studies in maize have shown that GCA effects can be predicted accurately using tools like GP. Consequently, implementing GP specially to predict GCA effects of the parental lines is a promising approach.

## 4. Materials and Methods

### 4.1. Selection of Parents, Hybrid Formation, and Trial Design

After screening >1100 lines for MLN under artificial inoculation [20], a set of 50 lines with varying level of MLN resistance or tolerance were chosen and crossed in a half diallel mating scheme and generated 307 F1 hybrids at the maize research station of Kenya Agriculture and Livestock Research Organization (KALRO), Kiboko, Kenya. All these hybrids with adequate seeds for evaluation over two seasons were harvested for this study. The inbred lines and hybrids were evaluated in one row 3m plots with two replicates in alpha lattice design for two seasons at MLN Screening Facility at the KALRO Research Center at Naivasha (Latitude 0°43′ S, longitude 36°26′ E, 1896 asl), Kenya. All the recommended standard agronomic practices were followed.

### 4.2. Viral Inoculum and Artificial Inoculation

The SCMV and MCMV isolates collected from MLN hotspot areas in Kenya were used to develop inoculum for this study. The isolates were confirmed by enzyme-linked immunosorbent assay (ELISA). Maintenance of inoculum purity was carried out on the susceptible hybrid H614 in separate greenhouses until artificial inoculation of entries in the field. MLN inoculum was prepared from an optimized combination of MCMV and SCMV viruses in the ratio of 1:4 to ensure uniform MLN pressure across the fields. Plant leaves used for inoculum were cut into small pieces and ground in 10 Mm Potassium phosphate at pH 7.0. The resulting sap extract was centrifuged at 12,000 rpm for two minutes and decanted with carborundum at 0.02 g/mL. Plants were inoculated at an inoculation spray pressure of 10 kg/cm^2^ using a backpack mist blower with an open nozzle of 2 inches in diameter. The presence of both viruses (SCMV and MCMV) in the inoculated field trials was confirmed by ELISA at approximately two weeks after inoculation. Each plot has 13 to 14 plants in each replications and disease data was recorded across all plants. MLN DS was visually scored on a plot basis for each replication, first score was started at three weeks after second inoculation (21 days post inoculation), afterwards at 10-day interval scoring was done for up to four observations (21-, 31-, 41-, and 51-days post inoculation). The scoring was done in the scale of 1 (no disease symptoms) to 9 (highly susceptible, complete plant death). After analyzing DS for each time score, the third score (41 days post-inoculation) was chosen for further analysis because of its higher heritability and full expression of disease symptoms. AUDPC, a quantitative measure of disease intensity with time was calculated for each plot using SAS 9.4 (SAS Institute Inc., 2015, Cary, NC, USA) to provide a measure of the progression of MLN severity across time. MLN Severity measurements over time were used to calculate the AUDPC by integrating the following formula for each parental line and hybrid separately:$$\mathrm{AUDPC}=\overset{n-1}{\underset{i}{\sum }}\frac{\left(X_{i}+X_{i+1}\right)\left(t_{i+1}-t_{i}\right)}{2}$$
where *X_i_* is an assessment of a disease severity (here in ordinal score of 1–9) at the *i*th observation, *t_i_* is time (in days) at the *i*th observation, and n is the total number of disease severity observations [47].

### 4.3. Phenotypic Data Analysis

Both lines and hybrids were evaluated in two seasons under artificial inoculation in Naivasha. For the analyses each season were treated as different environment. Since MLN DS was scored on an ordinal scale data was checked to know whether they follow all the applied statistical model assumptions like independent, normally distributed and having constant variance. These assumptions were checked by plotting residuals for each and across environments, which revealed MLN data met all the assumptions. In the process of quality check, we excluded the detected outliers from further analyses. Analysis of variance was determined for MLN DS and AUDPC within and across environments for inbred lines and hybrids by restricted maximum likelihood method using META-R (Multi Environment Trait Analysis R software) for calculating best linear unbiased predictions (BLUPs) for each lines and hybrids, and PROC MIXED procedure in SAS 9.4 (SAS Institute Inc., 2015, Cary, NC, USA) to calculate variance components, best linear unbiased estimates (BLUEs) and GCA effects. Dummy variables were used to separate genotypes into parental lines and hybrids. The phenotypic data of the parental inbred lines and hybrids were analyzed based on following linear model:$$y_{ijklm}=\mu +a+g_{ij}+l_{k}+{\left(gl\right)}_{ijk}+r_{lk}+b_{mlk}+e_{ijklm}$$
where *y_ijklm_* represents the phenotypic performance of *ij*th genotype (parental line *i* = *j*, or hybrid *i* ≠ *j*) in the *m*th incomplete block of the *l*th replication in the *k*th environment, *µ* is an intercept term, *a* is the group effect for lines and hybrids, *g_ij_* the genetic effect of the *ij*th genotype (parental line *i* = *j*, or hybrid *i* ≠ *j*), *l_k_* the effect of the kth environment, *(gl)_ijk_* the interaction of *ij*th genotype (parental line *i* = *j*, or hybrid *i* ≠ *j*) with *k*th environment, *r_lk_* the effect of the *l*th replication in the *k*th environment, *b_mlk_* the effect of the *m*th incomplete block in the *l*th replication of the *k*th environment, and *e_ijklm_* is the residual error term. Except *(gl)_ijk_* and *b_mlk_* all effects were treated as fixed to estimate BLUEs. Total variance components for lines and hybrids were estimated by assuming *l_k_*, and *r_lk_*, as fixed effect and all other effects as random.

The total variance of hybrids was further divided into variance due to GCA effects of males and females and SCA effects of crosses and their interactions. The mixed model used to estimate the variance components for hybrids was: $$y_{ijklm}=\mu +l_{k}+g_{i}^{\prime }+g_{j}^{\prime \prime }+s_{ij}+{\left(g^{\prime }l\right)}_{ik}+{\left(g^{\prime \prime }l\right)}_{jk}+{\left(sl\right)}_{ijk}+r_{lk}+b_{mlk}+e_{ijklm}$$
where *y_ijklm_* is the phenotypic performance of F1 hybrids in the *m*th incomplete block of the *l*th replication in the *k*th environment *l_k_* the effect of the kth environment, $g_{i}^{\prime }$ and $g_{j}^{\prime \prime }$ the GCA effect of the ith female line and jth male line, respectively, $s_{ij}$ is the SCA effect of crosses between lines *i* and *j*, ${\left(g^{\prime }l\right)}_{ik}\;and\;{\left(g^{\prime \prime }l\right)}_{jk}$ are GCA x environment effects of female and male lines, ${\left(sl\right)}_{ijk}$, the SCA x environment interaction effect and *e_ijklm_* is the residual error. Except environment and replication effect, all other effects were treated as random. A Wald’s *F* test [48] was used to test whether variances were significantly greater than zero. Broad sense heritability was estimated as the ratio of genotypic to phenotypic variance. Best linear unbiased estimate (BLUE) and best linear unbiased predictor (BLUP) for each parental line and hybrids were obtained for within and across environments. 

### 4.4. Molecular Data Analysis

DNA of all 50 inbred lines was extracted from 3–4 leaves old stage seedlings and genotyped using Genotyping by Sequencing (GBS) platform at the Institute for Genomic Diversity, Cornell University, Ithaca, USA as per the procedure described in earlier studies [49]. The GBS SNP datasets were filtered where a minor allele frequency of <0.10, heterozygosity of >5% and missing data rates >5% were excluded from further analysis in TASSEL ver 5.2 [50]. Finally, we used 13,450 SNPs for further analyses. Hybrid profiles were deduced from the parental fingerprints. Principal coordinate analyses (PCoA) was performed with Tassel ver 5.2 and then visualized in R software (http://www.R-project.org/).

### 4.5. Heterosis and Correlations

For each combination of parental lines, mid-parent value (MPV), relative mid-parent heterosis (MPH), and relative better parent heterosis (BPH) were calculated using hybrid performance (HP) as follows: MPV = (P1 + P2) / 2, MPH = [(HP − MPV) / MPV] × 100, and BPH = [(HP − P_max_) / P_max_] × 100, where P_max_ is the better resistant parent. Pairwise Pearson’s correlation coefficients (r) were calculated for HP with MPV, r(MPV:HP), and the GCA effects with the lines *per se* performance, r(GCA:*per se*). Further, we also tested the Pearson’s correlation of HP with the sum of GCA effects of both the parents, r(GCA: HP). The correlation between the Roger’s genetic distance between the parents and their MPH was estimated, r(GD:MPH). Correlation between Roger’s genetic distance and F1 HP was also calculated, r(GD:F1HP). All analyses were performed using the R software (http://www.R-project.org/).

### 4.6. Genomic Prediction 

Parametric prediction method G-BLUP accounting for the GCA was used to predict the performance of the single-crosses for MLN DS and AUDPC values. BLUEs across environments were used for the prediction. For the GP of the hybrid’s performance, we used a two-stage approach [13], where, in the first stage, the phenotypes were corrected for the experimental design effects, and in the second stage, the prediction models were fitted using the adjusted phenotypes. The details of the model and steps in implementation are explained in our earlier study [51]. In brief, this method is based on the commonly used additive effect model. Given the *n* by *p* matrix M describing the marker states of the *n* individuals at loci *p*, the additive model is defined as:*Y = mu + M* ∗ *beta + e*.

Here, *Y* is *n* by 1 the vector of phenotypic data, *mu* a fixed effect, *beta* a vector of marker effects and *e* a vector of errors. Note that the term GBLUP usually refers to a reformulated version using **g** = *M* * *beta*. In which **g** is assumed to be normally distributed with mean zero and variance covariance **G** that is the marker-derived relationship matrix obtained as VanRaden [52]. It must be highlighted that G matrix was obtained between parents and hybrids were predicted considering the GCA incidence matrix. For cross validations, the set of 307 hybrids were split into a training set (80%) and a testing set (20%) to test the predictive ability of the model; this process was repeated 50 times by random selection of the subsets. For each, correlation between predicted and observed values were calculated.

## Figures and Tables

**Figure 1 plants-09-00468-f001:**
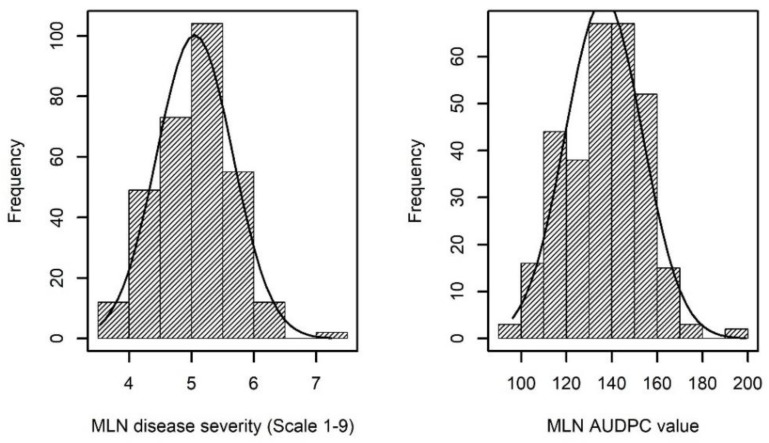
Phenotypic distribution for maize lethal necrosis (MLN) disease severity on the scale of 1–9 and the area under disease progress curve (AUDPC) values for 307 F1 hybrids.

**Figure 2 plants-09-00468-f002:**
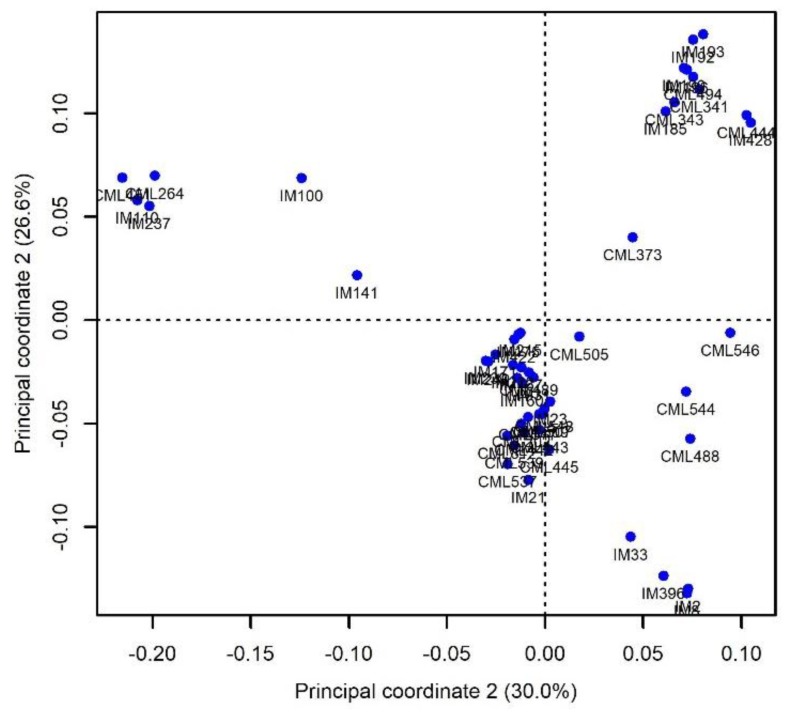
Principal coordinate analysis of the 50 parental lines based on Rogers’ distances. Values in parentheses refer to the proportion of variance explained by the principle coordinates.

**Figure 3 plants-09-00468-f003:**
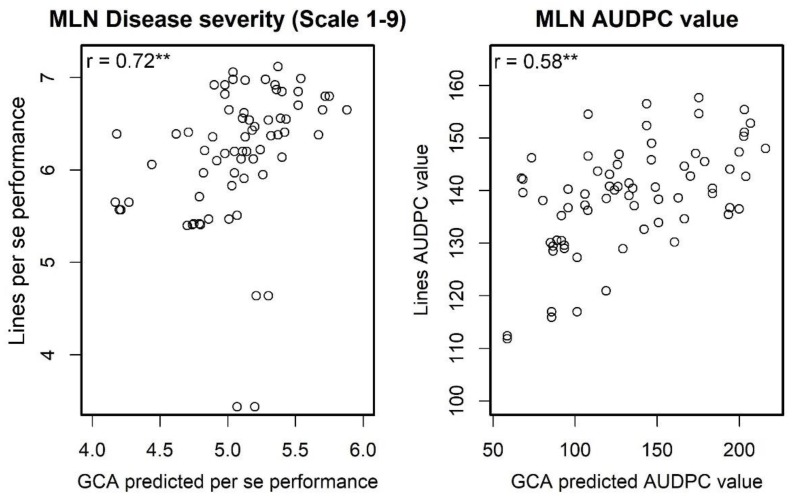
Association between line per se and general combining ability (GCA) for Disease severity and Area under disease progression curve (AUDPC). ** Means a significant correlation with *p*-value < 0.01.

**Figure 4 plants-09-00468-f004:**
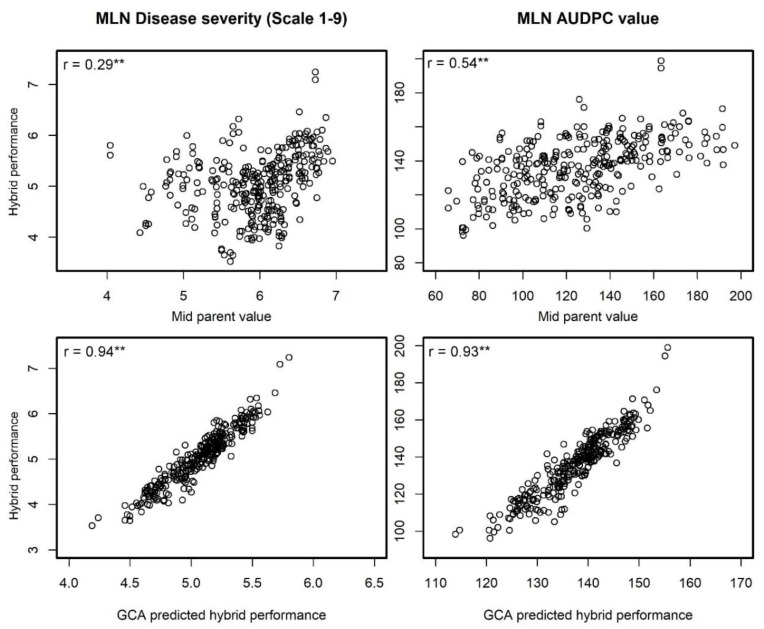
Association of mid parent value and GCA based prediction with observed F1 hybrid performance for disease severity and AUDPC. ** Means a significant correlation with *p*-value < 0.01.

**Figure 5 plants-09-00468-f005:**
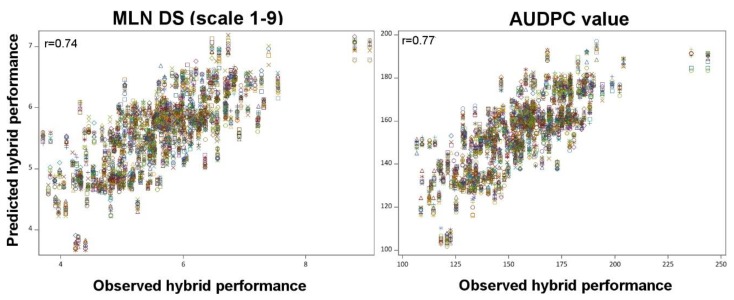
Association of marker based predicted and observed F1 hybrid performance for MLN disease severity (MLN DS) and AUDPC values in 50 random cross validation (80% to predict 20%). Each randomization is represented with a different color in the figure.

**Table 1 plants-09-00468-t001:** Estimates of mean, range, variance components (genotypic, σ^2^_G_; genotype x environment interactions, σ^2^_GxE;_ error, σ^2^_e_) and heritability (h^2^) for MLN disease severity (MLN DS) and area under disease progress curve (AUDPC) for parental lines and their 307 F1 hybrids.

Trait	MLN DS	AUDPC
Parents		
Mean	6.25	135.28
Range	3.44–7.12	58.69–216.05
σ^2^_G_	0.545 **	2303.16 **
σ^2^_GxE_	0.78 **	143.36 *
σ^2^_e_	1.40	1267.99
*h* ^2^	0.42	0.86
F1 Hybrids		
Mean	5.05	136.50
Range	3.53–7.24	96.30–199.10
σ^2^_G_	0.550 **	398.006 **
σ^2^_GCA (Female)_	0.156 **	114.85 **
σ^2^_GCA (Male)_	0.280 **	194.13 **
σ^2^_SCA_	0.114 **	89.02 **
σ^2^_GxE_	0.081 **	44.36 **
σ^2^_GCA (Female)xE_	0.030 **	18.47 **
σ^2^_GCA (Male)xE_	0.030 **	21.18 **
σ^2^_SCAxE_	0.030 **	11.07 **
σ^2^_e_	0.490	217.55
*h* ^2^	0.77	0.83

*, ** Significance at *p* < 0.05 and *p* < 0.01, respectively.

**Table 2 plants-09-00468-t002:** Mean and range of absolute and relative values of mid parent heterosis and better parent heterosis and the correlation between genetic distance of parental lines with F1 hybrids performance and mid parent heterosis for MLN disease severity and AUDPC values based on 307 F1 hybrids.

Heterosis		MLN DS	AUDPC
Absolute mid-parent heterosis	mean	−0.90	11.99
	range	−2.41–1.77	−53.96–68.36
Relative mid parent heterosis	mean	−14.57	14.19
	range	−38.63–43.83	−28.14–93.21
Absolute better parent heterosis	mean	−0.39	39.79
	range	−2.05–2.56	−45.78–98.50
Relative better parent heterosis	mean	−4.28	51.19
	range	−36.66–74.63	−24.93–100
r(GD:F1HP)	correlation	0.04	0.07
r(GD:MPH)	correlation	−0.11	−0.12

GD—genetic distance, MPH—mid parent heterosis.

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
