# Peer review of "Hybrid Breeding for MLN Resistance: Heterosis, Combining Ability, and Hybrid Prediction"

_plants, 2020, doi:10.3390/plants9040468_

Round 1

Reviewer 1 Report

The manuscript by Nyaga et al. aimed at comparing different predictors for MLN resistance of hybrids. Combining lines per se and hybrids evaluations and lines genotyping, they tested 4 predictors for MLN resistance and AUDPC area: the genetic distance between the parents, the mid-parental value, the GCA and the genomic prediction.

My main concern is that the manuscript is based on the results of statistical models that are not presented in the manuscript. There is a tentative in the 4.3 section but this paragraph must be deeply improved (for example the model of genomic prediction is absent) and must statistically rigorous and well written (that is clearly not the case).

Major points:

1/ The authors should be attentive to the use of “resistance” as sometimes in the manuscript, “tolerance” could be more appropriate (breeding for resistance but some lines are tolerant).

Phenotypes

2/ Authors should explain how the AUDPC was calculated. Is it the same model for all the genotypes? Whatever is the kinetics of the phenomena? Which mathematical adjustment was done ? Why ?

3/ The MLN DS phenotype is not clearly explained. Line 333: MLN DS was “scored”: how many plants / plot? 4 notations but is the attributed note a mean? A median? How do the authors managed with the plot replicates: means? lsmeans?

4/ the disease phenotypes are strongly interactive with environment so the phenotyping data should be re-analyzed in a statistically robust way.

Variance components

5/ Authors should draw inspiration from other articles presenting the decomposition of variance in hybrid designs with the GCA model – SCA. For example the Table 1 is not understandable without a statistical model clearly described.

The authors could find inspiration for example in

Petrizzelli et al. 2019, https://doi.org/10.1534/genetics.118.301635

or in

Seymour et al. 2016, www.pnas.org/cgi/doi/10.1073/pnas.1615268113

Predictors:

6/ The structuration of the material is not enough taken into account in the analyses of the results and in the discussion. The analysis of the genetic diversity and the link with hybrid performances should be stronger. A PCA including the lines and the hybrids or a correlation between Roger’s distance and the hybrid performances (as presented for the other predictors on figures 3,4 and 5) could be informative.

7/ The model of the GP and the experiment (number of replicated, size of the training set, cross-validation, etc.) should be presented. The accuracy could be an indicator of robustness. In the discussion section, line 291-293 the GBLUP-model is called but is written nowhere.

Minor points

Introduction

- Too many quotations of limited resources (lines 51, 53, 63,etc.)

- Line 63: the referenced article [17] is mainly cited for genomic prediction and not especially for limited resources

- Line 63: “types of hybrids” what does it means?

- Line 98: GP and MAS are not on the same plan. The first one is a way to predict the hybrid value; the second is a strategy for plant breeding. Maybe GP and QTL detection is better

Results

- Table 2: what is the definition of “range”? Maybe the calculation of the confidence interval or the standard deviation would be more informative. The notations are not consistent with these of the Materiel and Methods section 4.5

Figure 3: the outliers for the MLN severity should be discussed

Discussion

Line 256 “the complex nature of inheritance”. If complex means non-additive, this is not shown in the article (as the SCA is very low). So what complex means?

Mat & Meth.

Line 343: why SAS and R softwares? There are no explanations and no indications that justify their possible complementarity. Which software for which analysis?

Line 372: analyses were performed with ASReml-R: why? Do the authors need mixed models to calculated correlations?

Author Response

Comments and Suggestions for Authors

Reviewer 1

The manuscript by Nyaga et al. aimed at comparing different predictors for MLN resistance of hybrids. Combining lines per se and hybrids evaluations and lines genotyping, they tested 4 predictors for MLN resistance and AUDPC area: the genetic distance between the parents, the mid-parental value, the GCA and the genomic prediction.

Response: Thanks for the constructive comments on the study

My main concern is that the manuscript is based on the results of statistical models that are not presented in the manuscript. There is a tentative in the 4.3 section, but this paragraph must be deeply improved (for example the model of genomic prediction is absent) and must statistically rigorous and well written (that is clearly not the case).

Response: Thanks for your comment. In the revised manuscript we addressed all the comments, please see the responses below

Major points:

Comment1

1/ The authors should be attentive to the use of “resistance” as sometimes in the manuscript, “tolerance” could be more appropriate (breeding for resistance but some lines are tolerant).

Response: Thanks for your comment. In the revised manuscript we used the term resistance and tolerance appropriately as you suggested.

Comment 2

2/ Authors should explain how the AUDPC was calculated. Is it the same model for all the genotypes? Whatever is the kinetics of the phenomena? Which mathematical adjustment was done ? Why ?

Response: Thanks for your comment. We included the detail information about AUDPC in the revised manuscript, please see P12, L355-362.

Comment 3

3/ The MLN DS phenotype is not clearly explained. Line 333: MLN DS was “scored”: how many plants / plot? 4 notations but is the attributed note a mean? A median? How do the authors managed with the plot replicates: means? lsmeans?

Response: Thanks for your comment on data scoring. We included the detail information on each data scoring in the revised manuscript, please see P12, L348-352. In brief, we scored MLN on plot basis for each replication. Each plot has 13 plants, the score is average of 13 plants disease expression. Data was scored for 4 times at 10 days interval. Replicated data was plugged in to the SAS MLM model for estimating BLUEs, BLUPs and variance components.

Comment 4

4/ the disease phenotypes are strongly interactive with environment so the phenotyping data should be re-analyzed in a statistically robust way.

 Response: Thanks for your comment. We analyzed the data robustly with appropriate model. In the revised manuscript, we wrote in detail on the model we used, please see P12-13, L365-404.

Comment 5 - Variance components

5/ Authors should draw inspiration from other articles presenting the decomposition of variance in hybrid designs with the GCA model – SCA. For example the Table 1 is not understandable without a statistical model clearly described.The authors could find inspiration for example in Petrizzelli et al. 2019, https://doi.org/10.1534/genetics.118.301635 or in Seymour et al. 2016, www.pnas.org/cgi/doi/10.1073/pnas.1615268113

Response: We really appreciate reviewer suggestion and we agree with it. We read the suggested literature; our analyses were also similar. We used appropriate model, somehow, we were not explained the model in the original version of the manuscript, we rewrote the model used for analyses in the revised manuscript. Please see P12-13, L373-402.

Comment 6 - Predictors:

6/ The structuration of the material is not enough taken into account in the analyses of the results and in the discussion. The analysis of the genetic diversity and the link with hybrid performances should be stronger. A PCA including the lines and the hybrids or a correlation between Roger’s distance and the hybrid performances (as presented for the other predictors on figures 3,4 and 5) could be informative.

Response: Thanks for the good suggestion. In our results we included the prediction based on roger’s genetic distance with F1 hybrid performance and also with mid-parent heterosis, is presented in the Table 2. The parental lines used in this study are part of our previous association mapping study (Nyaga et al. 2020, https://doi.org/10.3390/genes11010016) Therefore we included limited information on parents diversity. Nevertheless, the correlations were calculated for GD with Fi performance and mid-parent heterosis, but the correlation was very poor. Therefore, we did not present the results in Figure format, presented in Table format.

Comment 7 - GP

7/ The model of the GP and the experiment (number of replicated, size of the training set, cross-validation, etc.) should be presented. The accuracy could be an indicator of robustness. In the discussion section, line 291-293 the GBLUP-model is called but is written nowhere.

Response: Thanks for the suggestion, and sorry for not presenting the GP section clearly in the manuscript. In the revised manuscript we gave details of the model and cross validations. Please see P14, L434-445.

Minor points

Comment 1 - Introduction

- Too many quotations of limited resources (lines 51, 53, 63,etc.)

Response: Thank you for pointing out this repeated word, we changed it in the revised manuscript

Comment 2

- Line 63: the referenced article [17] is mainly cited for genomic prediction and not especially for limited resources

Response: we deleted the less related reference in the revised manuscript.

Comment 3

- Line 68: “types of hybrids” what does it means?

Response: Types of hybrids refers to single cross or three-way hybrids. In Africa, still three-way hybrids are very popular. We clarified this in the revised manuscript.

Comment 4

- Line 98: GP and MAS are not on the same plan. The first one is a way to predict the hybrid value; the second is a strategy for plant breeding. Maybe GP and QTL detection is better

Response: Thanks for the suggestion, we included it in the revised manuscript.

Comment 5 - Results

- Table 2: what is the definition of “range”? Maybe the calculation of the confidence interval or the standard deviation would be more informative. The notations are not consistent with these of the Materiel and Methods section 4.5

Response: In Table 2, we provided the range of distribution for heterosis, as these values are calculated for 307 hybrids.

Comment 6

Figure 3: the outliers for the MLN severity should be discussed

Response: as for the suggestion, we discussed figure 3 results in discussion. Please see P10, L270-273.

Comment 7 - Discussion

Line 256 “the complex nature of inheritance”. If complex means non-additive, this is not shown in the article (as the SCA is very low). So what complex means?

Response: Thanks for the comment. Here we are referring complex nature because of two virus SCMV and MCMV involvement in causing MLN, as a result, not only genotypic effect and environmental effect cause disease severity, interaction between SCMV and MCMV also add to aggressiveness of disease spread. We clarified this in the discussion in the revised manuscript, please see P10, L262-266. “The interaction between two viruses SCMV and MCMV and their interaction with environments independently and/or together which is difficult to capture with current study may play a role here in the observed low correlations which warrants further study”.

Comment 8 - Mat & Meth.

Line 343: why SAS and R softwares? There are no explanations and no indications that justify their possible complementarity. Which software for which analysis?

Response: Sorry for not giving clarity on software uses. In the revised manuscript we clarify this in the materials and methods section in P12 L365-377. In brief, we used SAS to calculate BLUEs, VCs and GCA effects. META R to calculate BLUPs and R to calculate Heterosis, correlations and figures.

Comment 6

Line 372: analyses were performed with ASReml-R: why? Do the authors need mixed models to calculated correlations?

Response: Sorry for the mistake, we used R not ASREML R, we corrected this in the revised manuscript.

Reviewer 2 Report

The authors examined the MLN disease DS and AUDPC in F1 hybrids and their parental lines. Genetic distances of parental lines were examined and there was no clear relationship between GD and heterosis level. The authors showed correlation between genomic prediction of hybrid performance for MLN and the actual hybrid performance, suggesting that their methods are useful for the prediction of MLN resistance. Disease resistance is an important aspect in F1 hybrid cultivars and the prediction of this phenotype is useful for maize breeders. Thus, this manuscript provides useful information and is suitable for publication in this special issue. However, some concerns should be clarified in the revised manuscript.

  1. What is the relationship between MLN DS and AUDPC?
  2. Why did the authors mix the two viruses and did not inoculate each virus? Are there strains that are resistant for one virus but are susceptible for another virus ? Is the phenotype of SCMV and MCMV infection identical?

Author Response

Reviewer 2

Comments and Suggestions for Authors

The authors examined the MLN disease DS and AUDPC in F1 hybrids and their parental lines. Genetic distances of parental lines were examined and there was no clear relationship between GD and heterosis level. The authors showed correlation between genomic prediction of hybrid performance for MLN and the actual hybrid performance, suggesting that their methods are useful for the prediction of MLN resistance. Disease resistance is an important aspect in F1 hybrid cultivars, and the prediction of this phenotype is useful for maize breeders. Thus, this manuscript provides useful information and is suitable for publication in this special issue. However, some concerns should be clarified in the revised manuscript.

Response: Thanks for the constructive comments on the study. We followed your all suggestions and included in the revised manuscript.

Comment 1

What is the relationship between MLN DS and AUDPC?

Response: Thanks for the comment. MLN DS was scored at four times at 10 days intervals. Based on these MLN DS score, we calculated AUDPC values. MLN DS represents specific stage of disease expression, whereas MLN AUDPC represents across DS scores, across disease development stage/s. In the revised manuscript we included this information, please see P12, L348-359.

Comment 3

Why did the authors mix the two viruses and did not inoculate each virus? Are there strains that are resistant for one virus but are susceptible for another virus ? Is the phenotype of SCMV and MCMV infection identical?

Response: Thanks for the comment. Yes, it is true that some genotypes are good for SCMV and some other good for MCMV. It makes sense to screen them separately. However, for applied breeding, the final objective is to find the lines and hybrids which performs better for both the viruses. Screening separately for each virus doubles the cost of experiment and it misses the interaction between SCMV and MCMV which is very crucial in MLN disease development. For breeding purpose, we did for MLN screening, however, to understand the effect of each viruses, we are also screening the hybrids separately for SCMV and MCMV which will be our next research objective.

Round 2

Reviewer 1 Report

Just two notifications about the revised manuscript:

Line 396: The model used for the decomposition of variance in lines and hybrids need more precisions. The reviewer suggests:

- one model for the parental lines Y = mu + G+ E+ G*E

- one model for the hybrids as presented in the second version of the paper.

And the table 1 should be consistent with the models: as written by the authors, there is no sigma2G in hybrid model so why is there a value in the table?

The Y axis of figure 4: performance

Author Response

Reviewer comment

Line 396: The model used for the decomposition of variance in lines and hybrids need more precisions. The reviewer suggests:

- one model for the parental lines Y = mu + G+ E+ G*E

  • one model for the hybrids as presented in the second version of the paper 

And the table 1 should be consistent with the models: as written by the authors, there is no sigma2G in hybrid model so why is there a value in the table?

Response: Sorry for the silly mistake. Model 1 was used to estimate total variance components for parental lines and also for hybrids. Model 2 was used to decompose the total variance for hybrids to GCA variance for males, females and SCA variances. We clarified this in the table 1.

Comment 2

The Y axis of figure 4: performance

Response: Sorry for the mistake, we corrected the word in the revised figure.